# Genetic Landscape of Non-Remitting Neutropenia in Children and Chronic Idiopathic Neutropenia in Adults

**DOI:** 10.3390/ijms26146929

**Published:** 2025-07-18

**Authors:** Alice Grossi, Grigorios Tsaknakis, Francesca Rosamilia, Marta Rusmini, Paolo Uva, Isabella Ceccherini, Maria Carla Giarratana, Diego Vozzi, Irene Mavroudi, Carlo Dufour, Helen A. Papadaki, Francesca Fioredda

**Affiliations:** 1Laboratory of Genetics and Genomics of Rare Diseases, IRCCS Istituto Giannina Gaslini, 16147 Genova, Italy; alicegrossi@gaslini.org (A.G.); isabellaceccherini@gaslini.org (I.C.); 2Hemopoiesis Research Laboratory, School of Medicine, University of Crete, 71003 Heraklion, Greece; grigorios.tsaknakis@gmail.com (G.T.); i.mavroudi@uoc.gr (I.M.); e.papadaki@uoc.gr (H.A.P.); 3Clinical Bioinformatics, IRCCS Istituto Giannina Gaslini, 16147 Genova, Italy; francescarosamilia@gaslini.org (F.R.); martarusmini@gaslini.org (M.R.); paolouva@gaslini.org (P.U.); 4Haematology Unit, IRCCS Istituto Giannina Gaslini, 16147 Genova, Italy; mariacarlagiarratana@gaslini.org (M.C.G.); carlodufour@gaslini.org (C.D.); 5Genomics Facility, Italian Institute of Technology (IIT), 16163 Genova, Italy; diego.vozzi@iit.it; 6Department of Hematology, University Hospital of Heraklion, 71110 Crete, Greece

**Keywords:** non-remitting neutropenia, chronic idiopathic neutropenia, whole exome sequencing, genetic burden test

## Abstract

Non-remitting neutropenia in children and chronic idiopathic neutropenia (CIN) in adults have been described previously as peculiar subgroups of neutropenic patients carrying similar clinical and immunological features. The present collection comprising 25 subjects (16 adults and 9 children) mostly affected with mild (84%) and moderate (16%) neutropenia aimed to identify the underlying (possibly common) genetic background. The phenotype of these patients resemble the one described previously: no severe infections, presence of rheumathological signs, leukopenia in almost all patients and lymphocytopenia in one-third of the cohort. The pediatric patients did not share common genes with the adults, based on the results of the multisample test, while some singular variants in neutropenia potentially associated with immune dysregulation likely consistent with the phenotype were found. *SPINK5*, *RELA* and *CARD11* were retrieved and seem to be consistent with the clinical picture characterized by neutropenia associated to immune dysregulation. The enrichment and burden tests performed in comparison with a control group underline that the products of expression by the variants involved belong to the autoimmunity and immune regulation pathways (i.e., *SPINK5*, *PTPN22* and *PSMB9*). Even with the limitation of this study’s low number of patients, these results may suggest that non-remitting neutropenia and CIN in adults deserve deep genetic study and enlarged consideration in comparison with classical neutropenia.

## 1. Introduction

Children affected with non-remitting neutropenia, defined as late onset (LO) or long lasting (LL) neutropenia either with or without antibodies against neutrophils, have been classified within the provisional category of “likely Acquired Neutropenia”, according to the International Neutropenia Guidelines [1]. This group of patients shows a peculiar immunological pattern characterized by a reduction in absolute lymphocyte count and an immunophenotypic T- and B-cell profile similar to known autoimmune disorders, i.e., increased HLADR^+^ and CD3^+^ TCRγδ cells, reduced T-regulatory cells, increased CD27^−^IgD^−^ (double negative) naïve B-cells and a tendency toward reduced B-memory cells. In a minority of patients, pathogenic and likely pathogenic variants related to primary immuno-deregulatory disorders have been found by next generation sequencing (NGS) [2].

Adult patients with persistent and unexplained neutropenia who do not fulfil the diagnostic criteria of any underlying disease are characterized as chronic idiopathic neutropenia (CIN). A significant proportion of CIN patients also display an immune dysregulation pattern similar to LO/LL neutropenic children [3,4,5,6]. Specifically, CIN patients frequently display immunoglobulin disturbances, lymphopenia, activated T-lymphocytes with increased expression of HLA-DR and reduced memory B-cells [3,4,5,6]. The shared immunoregulatory disturbances between LO/LL neutropenia and CIN prompted us to investigate for a common genetic background between the two entities, testing the hypothesis that they may both belong to the same disease spectrum. We have, therefore, analyzed, by whole exome sequencing (WES), samples from pediatric LO/LL neutropenia and adult CIN patients, and the results are presented herein. To the best of our knowledge, no similar data are currently available in the literature.

## 2. Results

### 2.1. Immunological and Clinical Characteristics

The most important immunological characteristics are presented in Table 1. Leukopenia, defined as white blood cell counts below 4.0 × 10^9^/L in adults [7] and according to a different threshold for age in children, adolescents and young adults [8], was present in almost all patients (22/25; i.e., 88%), while lymphocytopenia, defined as counts below 1.5 × 10^9^ in adults [9] and below the lower limit according to age [8] in the younger group, affected 7/25 (i.e., 28%) patients. Based on the age-related reference values, none of the patients had decreased main class immunoglobulin levels. However, in accordance with the previously reported data, 5/25 (i.e., 20%) showed IgM above the threshold, and 3/25 (i.e., 12%), 7/25 (i.e., 28%) and 5/25 (i.e., 20%) patients displayed low IgG1, IgG3 and IgG4 subclass levels [10]. The most relevant reduction in lymphocyte subsets was related to natural killer cells (8/25, 32%), followed by depletion of CD19 in 3/25 subjects (12%). All patients had hemoglobin and platelet values within the normal range. As for the clinical/laboratory characteristics (Table 1), arthralgia without objective signs of arthritis was present in 7/25 (i.e., 28%) patients, fatigue was shown in 4/25 (i.e., 16%), mostly pediatric cases, and aphthae was referred by 4/25 (i.e., 16%) subjects. Two subjects had psoriasis, and one showed HLA-B27+ positivity with no symptoms/signs of ankylosis spondylitis. Anti-nuclear antibodies (ANA) were positive in 5/25 (i.e., 20%), antibodies against thyroid were present in 2/25 (i.e., 8%), and antibodies against neutrophils were present in 9/25 (i.e., 36%).

### 2.2. Genes Potentially Associated with Neutropenia

The results obtained through the multisample genetic analysis, the singleton analysis and the in silico panel of 538 genes associated with IEI (inborn error of immunity) (Figure 1) are summarized in Table 2 (for details, see Appendix A). Only genes known to be associated with the phenotype or potentially related to immunological disorders are reported.

Table 2 shows all genes shared by at least two patients within the entire cohort or present only in the pediatric or in the adult cohort, with variants characterized by a CADD (combined annotation dependent depletion) score greater than or equal to 15 and present in heterozygosity in a maximum of 10 controls from the Gaslini dataset. The genes carrying variants present only in a single subject, according to more stringent criteria such as a CADD score greater than or equal to 20, and variants not present in any healthy control in the Gaslini dataset are also shown. Lastly, the genes specifically associated with IEI but not emerged from the previous analyses because they (i) were not shared among patients, (ii) had a CADD score lower than 20 and (iii) were present in heterozygosity in not more than 10 controls from the Gaslini dataset are also presented. The Franklin database (https://franklin.genoox.com/clinical-db/home, accessed on 29 January 2025) was used to verify the prediction of each variant according to the ACMG criteria [11].

Overall, variants of uncertain significance (VUS), likely pathogenic (LP) variants and pathogenic (P) variants are reported. The Franklin database allows for the classification of VUS as either “hot” or “cold”, depending on whether the parameters used by the system lean towards a likely benign or LP variant, respectively. Hot VUS, represented in red according to this system, are “relevant”, having some probable roles in pathogenicity, while the cold VUS may not have any impact.

Hot VUS/LP/P variants that exhibit zygosity consistent with the gene’s inheritance pattern are highlighted in red. Furthermore, variants requiring special attention are marked in orange; these include (i) those affecting cancer predisposing genes, (ii) variants in genes involved in DNA repair or telomere maintenance, (iii) heterozygous variants in autosomal recessive (AR) inherited genes and (iv) variants in genes not yet associated with disease but potentially related to immune dysregulation. Finally, the cold VUS are indicated in grey.

In the following paragraphs, only the P/LP/hot VUS variants consistent with zygosity emerged through (i) the multisample test as for variants shared by children and adults, (ii) singleton analysis as for individual variants and (iii) genes belonging to an IEI panel by using a “more sensible” filter are described. The mechanisms of the disorders caused by the single variant and the consistency with the clinical features are extensively discussed.

#### 2.2.1. Multisample Analysis

A pathogenic homozygous variant in *SPINK5* (c.1302+4A>T) was found in Patient 10 (Pt#10), while Pt#4 is a heterozygous carrier of the *SPINK5* (p.R217*) variant (Table 2 and Appendix A). *SPINK5* is a serine protease inhibitor important for the anti-inflammatory and/or antimicrobial protection of mucous epithelia. It contributes to the integrity and protective barrier function of the skin by regulating the activity of defense-activating and desquamation-involved proteases. Interestingly, Pt#10 is affected by psoriasis (Table 1) [12].

Heterozygous *BRCA1* variants were seen in three patients, but one variant has a pathogenic prediction (LP) and was found in Pt#3 (p.D1692G) (Table 2 and Appendix A). A *BRCA1* mutation in a homozygous state is causative of Fanconi anemia (MIM #617883); otherwise, cancer predisposition is related to a single mutated allele. Breast cancer type 1 susceptibility protein plays a central role in DNA repair by facilitating cellular responses to DNA damage. Pt#10 has a family history of a very aggressive and precocious form of cancer. He inherited the mutation from his father who has been affected by prostatic carcinoma very early in life [13].

Two *MSTIR* variants, c.1880+2T>G and p.V323Wfs*15, classified as two hot VUS, were detected in Pt#21 and Pt#1, respectively (Table 2 and Appendix A). Macrophage-stimulating protein receptor alpha chain *(MSTIR)* regulates many physiological processes by binding to the MST1 ligand, including cell survival, migration and differentiation. Ligand binding at the cell surface induces autophosphorylation of RON (alias for MSTIR) on its intracellular domain that provides docking sites for downstream signaling molecules. Following activation by the ligand, MSTIR interacts with the PI3-kinase subunit PIK3R1, PLCG1 or the adapter GAB1 [14].

#### 2.2.2. Singleton Analysis

This analysis showed two hot VUS and one LP variant only in the adult cohort, affecting the *GATA2*, *GJB4* and *ITGB3* genes, with a zygosity consistent with the expected inheritance (Table 2 and Appendix A).

In detail, the *GATA2* (p.H127N) hot VUS was identified in Pt#15. The patient has mild neutropenia discovered in a routine checkup and remains asymptomatic to date. She also screened as HLA-B27 positive but with the absence of any musculoskeletal symptoms. The GATA transcription factors have emerged as candidate regulators of gene expression in hematopoietic cells, usually dendritic cells, monocytes, B lymphocytes and NK lymphocytes. Interestingly, this patient had lymphopenia with particularly low NK cells [15,16].

*GATA2* is involved in familial leukemia and in a complex congenital immunodeficiency that evolves over decades and leads to predisposition to infection and myeloid malignancy. The patient is not showing any signs of dysplasia or any susceptibility to infections to date.

The *GJB4* hot VUS (p.I30T) was identified in Pt#20. Erythrokeratodermia variabilis et progressive (MIM #617524) is the phenotype associated with the presence of a causative variant in this gene, characterized by persistent plaque-like or generalized hyperkeratosis and transient red patches of variable size, shape and location. The patient has had mild neutropenia for the last 35 years and does not show any signs of skin abnormities or infections.

The *ITGB3* (p.A754T) LP variant was detected in Pt#23. The *ITGB3* gene encodes glycoprotein IIIa (GP IIIa), the beta subunit of the platelet membrane adhesive protein receptor complex GP IIb/IIIa. The mutation in this gene is responsible for a bleeding disorder, platelet-type 24 and Glanzmann thrombasthenia 2 (MIM #607759), which is a feature not consistent with neutropenia. Indeed, the patient is suffering from mild neutropenia, with no major infections and diffuse bone pain and myalgia [17,18].

#### 2.2.3. Inborn Error of Immunity Panel Analysis

According to the 538 gene custom panel compiled from the International Union of Immunological Societies (IUIS) Expert Committee and PanelApp England (Primary immunodeficiency or monogenic inflammatory bowel disease (Version 4.0)), two hot VUS and one P variant were found (Table 2 and Appendix A): * TGFBR2 *(p.R224H) in Pt#2; *CARD11* (p.S1026C) in Pt#25; *MEFV* (p.M694V) in Pt#20.

The hot VUS of *TGFBR2* (p.R224H), which is basically associated with Loeys–Dietz syndrome (MIM #610168), has no clinical consistency in Pt#2. On the contrary, heterozygous VUS in the *RELA* gene (p.S536del), carried again by Pt#2, consistently causes familial Bechet-like autoinflammatory disease-3 (MIM #618287). Patients’ ulcerations occur specifically in the oral, gastrointestinal and vaginal mucosa rich in microbiota, suggesting they may have served as inflammatory stimuli inducing TNF release, resulting in mucosal injury and impaired epithelial recovery [19]. The symptoms of Pt#2 fit with this description.

Pt#25 carried an LP variant (p.S1026C) in *CARD11*. The gene encodes a protein that acts as a scaffold for nuclear factor kappa-B (NF-κB) activity in the adaptive immune response controlling peripheral B-cell differentiation and a variety of critical T-cell effector functions. *CARD11* is associated with immunodeficiency (MIM #617638/615206/616452). Indeed, the patient displays reduced levels (borderline) of IgG1 and IgG3 subclasses.

*MEFV* is the gene of familial Mediterranean fever (FMF) (IM #134610/249100), one of the most common monogenic autoinflammatory diseases. Pt#20 carried a P variant (p.M694V) in *MEFV*. This gene is involved in inflammatory reactions through altered leukocyte apoptosis, secretion of interleukin-1beta (IL-1β) and activation of the NF-κB pathway, and, thereby, the degree of inflammation. The patient had no features of clinical FMF and was just suffering from mild neutropenia [20].

### 2.3. Gene Clustering

In Appendix A, all genes resulting from the previous analyses are collected, and the contribution of the variants found in each gene is highlighted with different colors according to the pathogenicity prediction described by Franklin. The genes are divided according to the functional pathway they belong to using EnrichR and Reactome Pathways 2024 (https://maayanlab.cloud/Enrichr/, accessed on 11 July 2025). The categorization of the genes revealed a diverse array of biological functions, highlighting the complexity of cellular processes that may be associated with the neutropenia phenotype. Prominent among these are pathways related to the innate immune system and adaptive immune system (response and inflammation), suggesting a potential strong link between immune dysregulation and neutropenia occurrence. Additionally, a significant portion of the genes contribute to DNA repair and genome stability. Overall, this classification underscores the diverse roles these genes may have in contributing to the neutropenia phenotype (Figure 2).

#### Multisample Analysis of Total Patient Group vs. Healthy Controls

The multisample dataset, including both neutropenic patients (n = 25) and healthy controls (n = 75), was used to assess the enrichment of rare, deleterious, hot VUS, LP, and P variants. Several comparisons were performed, including LL/LO patients versus healthy controls, CIN patients versus healthy controls, and total neutropenia patients (i.e., LL/LO plus CIN) versus healthy controls. In this latter test, *SPINK5* almost reached the nominal p threshold (*p*-value: 0.06), suggesting a potential burden of pathogenic variants. Additionally, in the CIN versus healthy controls comparison, *RAD50* exhibited a significant enrichment after multiple testing correction (*p*-value: 0.019), indicating an enrichment of suggestive variants in this gene. The same vcf file was used for testing rare variant association, focusing on rare and deleterious variants with high, moderate or modifier impact on transcripts. Using RVTEST, different statistical models were applied: (1) the CMC model (Burden test) considers variants as if they contribute equally, and in this particular model, all variants are collapsed into a single score; (2) for the SKAT analysis, the variants are analyzed individually, with weights assigned based on MAF (rarer variants have higher weights); (3) the SKAT-O approach is considered the optimal test, combining SKAT and Burden. If all variants have a similar effect (all deleterious or all protective), it behaves like Burden. If variants have mixed effects (some increase and some decreased risk), it behaves like SKAT.

Among these analyses, *PTPN22* and *PSMB9* showed a significant enrichment of rare variants using the CMC model (Appendix A), even after multiple testing correction (Benjamini–Hochberg method). Additionally, MYOF reached an FDR < 0.05 in the SKATO analysis (Appendix A).

Overall, 23 genes from the 538 genes tested in the custom panel showed at least one FDR-adjusted *p*-value ≤ 0.08 between the two approaches. These genes were further categorized into four key biological processes: cellular growth and cellular development, inflammation and immunological regulation, cellular metabolism and cellular transport (Figure 3).

Among the significant enriched genes, *PTPN22* is a lymphoid-specific intracellular phosphatase that interacts with the adaptor protein CBL and plays a role in regulating T-cell receptor signaling. The gene *PSMB9* is particularly important for inflammation and immunoregulatory pathways. Notably, this latter gene, located in the MHC class II region, encodes a beta subunit of the 20S core of the immunoproteasome, whose expression is induced by interferon-gamma and is involved in antigen processing. Finally, *MYOF* encodes a type II membrane protein structurally related to dysferlin, involved in membrane repair and regeneration through calcium-mediated fusion events, with mutations leading to muscle weakness affecting both proximal and distal muscles.

## 3. Discussion

CIN in adults and LL/LO neutropenia in children and adolescents, respectively, are two categories of neutropenic patients who seem to have peculiar clinical and biochemical features when compared to other forms of neutropenia, either acquired or congenital. In these patients, previous studies identified a subgroup whose immune dysregulation background was sustained by the presence of pathogenic variants of IEI [2,3,4,5]. The hypothesis is that neutropenia may not represent “the” disease but rather an epiphenomenon of immune dysregulation.

In this perspective, the application of WES analysis intended to look for any common genetic background in CIN and in LO/LL able to justify a common “root” of these entities. The immunological and clinical characteristic of the population described in the present paper basically resembled those described previously [2,3,4,5], thus delineating again the different composition of these groups compared to the “classical neutropenia” categories; this was further confirmed by the absence of any common neutropenia variants.

The genes found with the multisample test using “less stringent” parameters than the ones applied in the IEI panel were *SPINK5, BRCA1* and *MSTIR1*. *SPINK5* is related to a skin disorder and is consistent with the clinical picture of Pt#5, who is affected with psoriasis; the possible connection with neutropenia might be explained by the expected reduced expression of LEF1, which in turn deregulates lymphocyte activity.

*SPINK5* is a serine protease inhibitor that plays a pivotal role in a wide variety of immune and inflammatory processes, including T- and B-cell differentiation, activation of cytokines and complement and recruitment of inflammatory cells. It is difficult to say whether neutropenia may be the result of such mechanisms [21].

Pt#3 was found to be a carrier of *BRCA1*; indeed, a high incidence of early and severe cancer in this patient’s family has been documented. His father, affected with an aggressive form of prostatic cancer, did carry the same mutation. As for the *MSTIR* gene, only hot VUS have been found; being that this gene is involved in cellular traffic, it could be hypothesized that neutropenia could be the result of neutrophil migration, but again, this remains only a hypothesis. As for the singleton analysis, *GATA2*, *GJB4* and *ITGB3* mutations were identified. *GATA2* is a transcription factor that plays an essential role in the development and proliferation of hematopoietic and endocrine cell lineages. Pathogenic variants in this gene cause autosomal-dominant GATA2 deficiency, which may present clinically as Emberger syndrome, immunodeficiency 21, susceptibility to acute myeloid leukemia or myelodysplastic syndrome, WILD syndrome (warts, immunodeficiency, lymphedema), MonoMAC or DCML (dendritic cells, monocytopenia and lymphopenia) deficiency. *GATA2* germline mutations are also reported in patients with myeloid malignancy without evidence of preceding immunodeficiency or other disturbances [22].

In our patient (Pt#15), we detected the *GATA2* (p.H127N) variant, a missense mutation affecting codon 127 that is predicted to be deleterious and has not been reported in GnomAD before. The patient is not showing any GATA2-related clinical signs. Regarding the hot VUS GJB4 (p.I30T) variant identified in Pt#20, an association with the neutropenia phenotype cannot be established. The same applies for the *ITGB3* (p.A754T) LP variant detected in Pt#23. When the panel of IEI is studied with a high sensitivity, some interesting data are found, at least for *RELA* and *CARD11*, showing more consistent connection with the neutropenia feature.

*RELA* haploinsufficiency causes an autosomal-dominant, mucocutaneous disease named autoinflammatory disease, familial, Bechet-like-3. This disorder provides a counterpoint to autoinflammatory diseases characterized by increased inflammatory cytokine signaling. Defects in the genes encoding the NF-κB regulatory proteins result in NF-κB overactivation and multisystemic autoinflammation characterized by oral ulcers, arthritis, uveitis and/or vasculitis [23,24]. The history of Pt#2 began as isolated chronic mild symptomatic autoimmune neutropenia with occasional mouth ulcers. At the age of 17 years, the patient developed frank arthritis, which required anti-TNF therapy. Autoimmune cytopenia has been reported in patients carrying this variant too [24].

*CARD11* is a membrane protein acting as a key signaling scaffold that controls antigen-induced lymphocyte activation during the adaptive immune response (NF-κB, JNK and mTOR pathway). Germline *CARD11* mutations can result in gain/loss of function of the protein, leading to different phenotypes (SCID, BENTA disease, atopy, CVID). Several heterozygous hypomorphic/dominant negative variants of *CARD11* have demonstrated a loss of protein function, exhibiting high penetrance and variable expressivity [25,26]. The largest heterozygous germline loss-of-function case series is composed of 48 patients with a broad phenotype, including atopy, viral infections of the skin and respiratory tract, hypogammaglobulinemia and autoimmunity. Nonetheless, 14% of this cohort were neutropenic [27]. In the case of our patient, neutropenia is chronic and asymptomatic, and the patient has no history of infections and other immunodeficiency-related clinical signs apart from slightly reduced IgG1 and IgG3 levels.

Pt#20 was a heterozygous carrier of the *MEFV* gene (p.M694V) encoding pyrin, which functions as an innate immune sensor that can trigger the formation of an inflammasome, allowing for the production of inflammatory mediators during infection [28].

The p.M694V mutation is very common in the Mediterranean population, probably due to the selective advantage based on heightening the inflammatory response to some pathogens endemic in this geographic region [29,30]. The clinical spectrum of p.M694V carriers is wide, from a silent phenotype (mainly in heterozygous individuals) to severe forms (i.e., amyloidosis); several reports outline that fever, chest pain and abdominal pain are the main presentation symptoms shown either from homozygous or heterozygous carriers. Cytopenia and, in particular, neutropenia are not described as associated factors; probably, this mutation is not consistent with the actual picture. Indeed, the patient does not demonstrate any typical clinical manifestations associated with FMF.

The lack of shared genes between the adults and the pediatric population may be due to the small size of our sample. Moreover, the direct correspondence between the LP/P variants and the role of a number of hot VUS in the clinical phenotype have to be better defined to understand the real weight in the clinical settings.

We can hypothesize that the neutropenia phenotype may result from the combined effects of multiple genetic variants rather than a single causative mutation, thus considering that these disorders are rather far from the classical monogenic disorders. This concept, known as oligogenic or polygenic inheritance, is increasingly recognized in complex diseases such as ulcerative colitis [31,32]. In this view, the pathway enrichment analysis, utilizing tools like EnrichR and Reactome Pathways 2024 and including all the genes being mutated in our cohort of patients, was performed.

This analysis highlighted the involvement of key biological functions in neutropenia. Interestingly, the enrichment of pathways related to the innate and adaptive immune systems seems to suggest that immune dysregulation plays a critical role in the development of this condition. Furthermore, the identification of genes involved in DNA repair and genome stability indicates that genomic integrity may also be compromised in neutropenia patients. This categorization of affected pathways provides valuable insights into the cellular mechanisms that may be disrupted in the disease and underlines the importance of the longitudinal follow-up.

In the search for candidate genes, we also performed comparisons between neutropenia patients and healthy controls as well as rare variant association testing. The identification of *SPINK5* and *RAD50* as potentially significant suggested their involvement in the LL/LO and CIN groups, respectively. Furthermore, the rare variant association testing, using models like CMC, SKAT and SKAT-O, revealed additional genes, such as *PTPN22, PSMB9* and *MYOF*, that may contribute to the disease phenotype. These findings underscore the utility of SKAT-O and CMC in uncovering rare variant associations within biologically relevant genes, particularly in complex immune-mediated conditions. The convergence of immune regulatory pathways and membrane dynamics observed here warrants further investigation, potentially offering novel mechanistic insights.

Moreover, the fact that 23 genes showed some level of significance across these approaches underscores the complexity of the genetic landscape of neutropenia. Despite the heterogeneity of the genetic variants identified, many of the implicated genes are involved in the regulation of immune responses.

This observation supports the hypothesis that immune dysregulation may play a crucial role in the pathogenesis of neutropenia.

## 4. Materials and Methods

The WES analysis was carried out according to the multisample method, combining, in a single variant call format (vcf) file, the data of 9 pediatric patients collected at the Hematology Giannina Gaslini Institute (Genoa, Italy) and 16 adult patients selected at the Department of Hematology, University Hospital of Heraklion (Crete, Greece). Variants reported in GnomAD (https://gnomad.broadinstitute.org/) v3.1.2 with an allele frequency equal to or lower than 5% or never seen before and with a combined annotation dependent depletion (CADD) score v1.6 (https://cadd.gs.washington.edu/) equal to or greater than 15 were selected. An internal database of the Giannina Gaslini Institute, consisting of over 7000 WES of pediatric patients affected with various disorders and their healthy parents, was used to verify the possible presence and consequent frequency of these variants also in the reported healthy adult population. It was arbitrarily decided to still consider variants present in heterozygosity in up to 10 healthy controls. In this multisample analysis, coding region variants and splicing variants shared by at least two patients were selected, regardless of whether the variants were identical or distinct but located within the same gene. Applying these criteria, we looked at mutant genes and variants shared between both the pediatric and adult cohorts, only in the pediatric cohort and only in the adult cohort. Subsequently, the WES data from the 25 patients of this study were analyzed individually (singleton analysis) using more stringent criteria: unique variants with a CADD score equal to or greater than 20 and absent in the internal database of healthy controls.

Finally, an in silico custom panel of 538 genes associated with inborn errors of immunity (IEI) compiled from the International Union of Immunological Societies (IUIS) Expert Committee [33] and PanelApp England (Primary immunodeficiency or monogenic inflammatory bowel disease, Version 4.0) (https://panelapp.genomicsengland.co.uk, accessed on 11 July 2025) was applied to the multisample file, looking for variants present in a single patient, with CADD more than or equal to 15 and more likely associated with the phenotype.

A case-control approach was also adopted comparing the total patient group with a control group consisting of 75 healthy individuals (median age 62 years, range 52–71 years) from the cohort described above. This cohort included patients with mild COVID-19 and no need for hospitalization or advanced respiratory support. Clinical and laboratory parameters did not indicate either leukopenia or neutropenia.

Starting from this enlarged dataset, two types of analyses were conducted: (1) a Fisher exact test corrected for false discovery rate (FDR) to compare the total patient group (LL/LO plus CIN) with healthy controls, testing the presence and enrichment of rare, deleterious, hot VUS, likely pathogenic and pathogenic variants; (2) a rare variant association test by means of the RVTESTS software (version: 20181226) [34] using the silico gene panel composed of 538 associated genes cited above and considering rare variants with minor allele frequency (MAF) less than 0.01 in the study cohort; CADD ≥ 20 or without CADD; and high, moderate or modifier impact. The variants were first annotated with ANNO (github.com/zhanxw/anno, accessed on 11 July 2025) and then analyzed using different tests: Kernel method (SKAT and SKAT-O), Burden test (ExactCMC) and variable threshold method (Price model). The significance threshold was determined by applying the FDR correction.

## 5. Conclusions

Even with the limits of the present collection, mainly related to the low numbers of enrolled patients whose ancestry was not extensively collected, the working hypothesis of neutropenia as a result of genetically driven disimmunity takes shape.

The genetic landscape of LO/LL and CIN neutropenias is complex, involving a combination of rare and common variants in genes mainly involved in immune regulation.

The present findings suggest that these conditions are not classical monogenic disorders but rather complex diseases resulting from the interplay of multiple genetic and environmental factors.

All the premises are the basis to design future goals aimed, firstly, at increasing the sample size to find out shared variants and to analyze any possible epigenetic/epigenomic influence. Recognizing specific patterns of immune dysregulation emerged as neutropenia, interpreting complex genetic testing and choosing any possible target treatment can be challenging but might definitely change the future of the affected subjects.

## Figures and Tables

**Figure 1 ijms-26-06929-f001:**
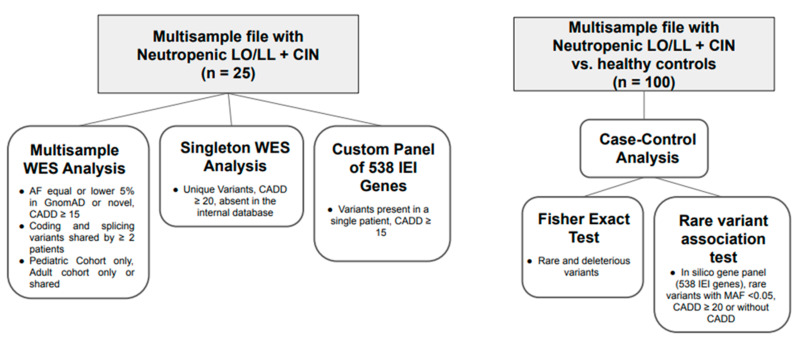
WES analysis pipeline. The flowchart summarizes the WES analysis pipeline used in this study. It begins with multisample variant calling from pediatric and adult cohorts, followed by filtering based on population frequency and deleteriousness (CADD score). Variants were compared across groups (shared, pediatric only, adult only), with additional singleton analysis applying stricter criteria. A custom gene panel (538 IEl-related genes) was then used to identify potentially pathogenic variants. Finally, a case-control approach was applied, incorporating statistical association tests (Fisher’s exact test, SKAT, SKAT-o, Cm C, VT) to detect gene-level enrichment of rare damaging variants in patients versus healthy controls.

**Figure 2 ijms-26-06929-f002:**
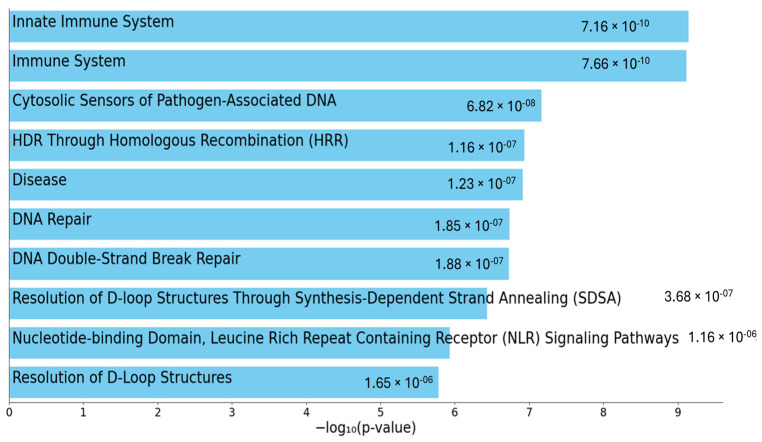
The top 10 enriched terms for the input gene set are displayed based on the −log_10_(*p*-value), with the actual *p*-value shown next to each term.

**Figure 3 ijms-26-06929-f003:**
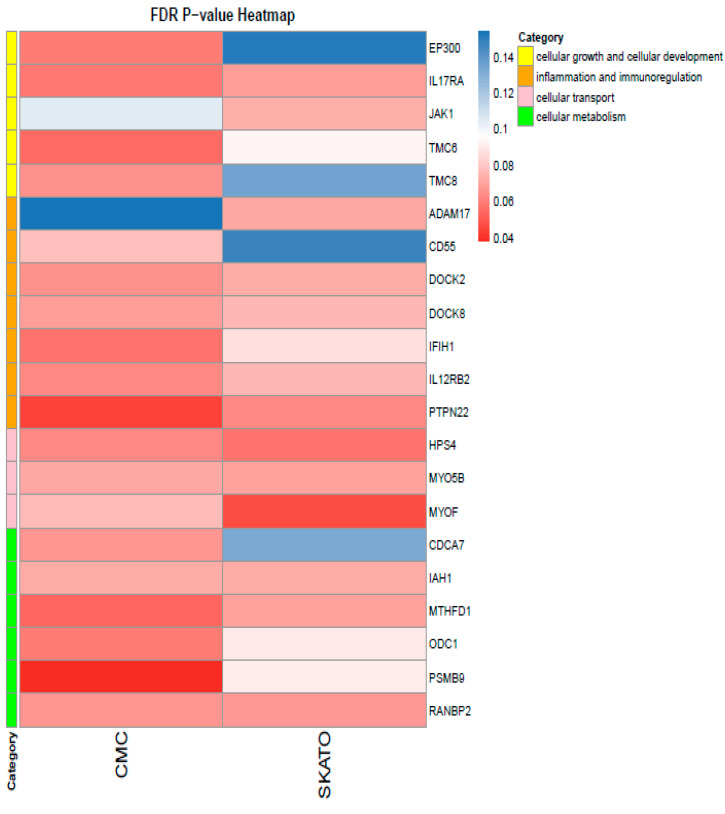
Heatmap showing genes identified as slightly significant (FDR-corrected *p*-value between 0.08 and 0.05, light red) and significant (FDR-corrected *p*-value < 0.05, dark red). The columns indicate, with corresponding colors, genes identified as nominally significant or significant using the CMC and SKAT approaches, as described in the Materials and Methods Section 4. The legend categorizes these genes into broader functional groups.

**Table 1 ijms-26-06929-t001:** Clinical and hemato/immunological features of the whole cohort.

PT	Gender	Ethnicity	Consanguinity	Date of Birth	Age at Onset (y)	Age at Diagnosis of NP (y)	Duration FUP (mo)	Main Clinical	Family History	AutoimmunityIncluding AbAN	WBC (×10^9^/L)	Neu (×10^9^/L)	Lymph (×10^9^/L)	Mono (×10^9^/L)	IgA (48–368 mg/dL)	IgG (701–1600 mg/dL)	IgM (25–170 mg/dL)	IgG1 (490–1140 mg/dL)	IgG2 (150–640 mg/dL)	IgG3 (20–110 mg/dL)	IgG4 (8–140 mg/dL)	CD3%	CD3/CD4%	CD3/CD8%	CD19%	CD3-(CD16/56)+%
Pt1	F	C *	N°	1991	16	27	41	Fatigue/Apthae/ Arthralgias	NR #	ANA+ Ab AN+	2.3	0.5	1.3	0.3	153	1171	193	752	369	38.4	9.2	82.7	50.3	18.4	11.6	4.0
Pt2	F	C *	N°	2003	2	16	32	Fatigue/Apthae/ Arthralgias	NR #	Ab AN+	1.7	0.5	0.97	0.15	220	1028	67	546	435	19.3	5.3	84.9	57.0	20.1	5.4	8.3
Pt3	M	C *	N°	2002	12	13	72	Apthae	Early Cancer ^a^	ANA+ Ab AN+	3.5	0.4	2.2	0.3	222	1196	156	949	328	109	84	78.0	45.0	27.7	8.6	13.4
Pt4	F	C *	N°	2003	11	13	66	Fatigue/Apthae	Neurodegenerative disease ^b^	ANA+ Ab AN+	2.7	0.45	1.6	0.4	136	1436	305	881	437	37.2	112	78.0	58.7	2.4	14.3	5.0
Pt5	M	C *	N°	2011	2	4	64	Nothing	NR #	Ab AN+	2.9	0.95	1.5	0.24	116	1207	67	1060	189	74	80	77.4	43.4	25.2	12.1	6.6
Pt6	F	C *	N°	2015	0	0.3	66	Nothing	Early Cancer ^c^	Ab AN+	4.8	0.48	1.9	0.35	54	905	75	677	221	36	24	67.0	39.0	19.0	11.0	18.4
Pt7	F	C *	N°	2001	17	19	12	Nothing	NR #	Anti Thy Ab+	2.2	0.32	1.4	0.4	58	1322	223	873	313	37.1	12.2	82.1	54.4	20.8	13.0	4.0
Pt8	F	C *	N°	2000	17	18	41	Nothing	Cancer in the family ^d^	Ab AN+	1.9	0.22	1.1	0.4	185	1308	68	748	692	52	12	78.4	39.5	22.3	12.0	5.6
Pt9	M	C *	N°	2003	15	15	34	Fatigue	NR #	Anti Thy Ab+ Ab AN+	1.9	0.35	1.1	0.3	165	959	75	750	189	22	118	81.3	34.4	31.1	9.9	6.9
Pt10	M	C *	N°	1993	25	26	48	Mild psoriasis	Neutropenic sibling	Ab AN+	2.9	0.8	1.3	0.6	245	1340	166	785	417	29.2	19.2	76.5	48.0	25.3	14.7	7.8
Pt11	F	C *	N°	1968	50	50	60	Arthralgias	NR #	Neg	3.1	1.5	1.1	0.4	117	1230	130	822	393	52.1	0.7	66.3	44.0	19.4	16.6	16.8
Pt12	F	C *	N°	1964	36	38	240	Arthralgias	NR #	Neg	3.0	1.2	1.5	0.3	144	1040	82	615	391	25.4	2.5	68.6	52.0	15.0	14.4	14.3
Pt13	F	C *	N°	1979	18	18	372	Arthralgias	Neutropenic sibling	Neg	3.3	1.2	1.6	0.3	253	1100	135	651	598	34.0	41.1	71.0	38.0			
Pt14	M	C *	N°	1957	32	32	156	Nothing	NR #	Neg	2.8	0.7	1.4	0.5	307	720	45	367	355	13.5	35.6	74.7	30.9	41.2	9.6	14.8
Pt15	F	C *	N°	1963	54	56	36	Ankylosis spondylitis	NR #	Neg	2.4	1.1	1.0	0.3	222	930	187	610	340	35.3	48.4	79.7	52.4	22.5	11.6	8.8
Pt16	F	C *	N°	1951	35	35	180	Arthralgias	NR #	ANA+	3.3	1.3	1.5	0.4	189	1330	37	925	226	67.4	3.0	78.0	58.2	23.8	6.4	
Pt17	M	C *	N°	1973	27	27	209	Nothing	NR #	Neg	3.2	0.5	2.0	0.4	252	1340	647	703	570	31.0	57.5	79.7	38.9	38.5	13.6	5.3
Pt18	M	C *	N°	1973	26	30	96	Psoriasis	NR #	Neg	2.7	1.4	1.0	0.3	265	1420	63	895	436	13.9	50.0	51.2	29.2	26.5	10.7	28.2
Pt19	F	C *	N°	1953	60	66	60	Nothing	NR #	Neg	2.5	0.4	1.5	0.5	120	1380	79	949	423	8.3	108.0	65.0	39.0	35.0	16.3	13.7
Pt20	F	C *	N°	1954	35	35	276	Nothing	NR #	ANA+	3.8	1.6	1.7	0.4	381	1120	48	872	185	18.1	0.3	71.0	37.0	34.0	18.0	8.0
Pt21	F	C *	N°	1968	32	32	288	Nothing	NR #	Neg	3.7	1.5	1.7	0.3	155	1200	193	717	260	24.0	107.0					
Pt22	F	C *	N°	1958	23	25	276	Nothing	NR #	Neg	4.0	1.5	1.6	0.5	145	790	120	363	370	16.7	16.9	72.2	44.2	27.7	16.3	11.4
Pt23	F	C *	N°	1968	41	50	72	Arthralgias	NR #	Neg	3.0	1.0	1.6	0.3	244	981	121	630	388	16.3	94.7	63.8	35.0	23.0	12.3	13.8
Pt24	M	C *	N°	1999	18	18	120	Nothing	NR	Neg	2.8	1.0	1.5	0.2	62	1150	183	721	269	28.7	59.4	77.0	57.4	19.6	8.1	
Pt25	F	C *	N°	1966	40	40	216	Nothing	NR	Neg	4.0	1.6	1.8	0.5	104	836	29	440	201	19.1	108.0	84.9	54.9	28.8	9.1	11.7

In light grey (from Pt1 to Pt9) is highlighted the pediatric population. Ethnicity: C * = Caucasian, Consanguinity: N° = Non-consanguineous, # NR = Not reported. Cancer in detail: (^a^) Early prostate cancer (father); (^b^) Neurodegenerative disease (sister); (^c^) Early Hodgkin disease (mother); (^d^) Pediatric acute lymphoblastic leukemia (first cousin). WBCs, white blood cells; ANA, anti-nuclear antibodies; Anti Thy Ab, antibodies against thyroid; Ab AN, anti-neutrophil antibodies.

**Table 2 ijms-26-06929-t002:** Overview of all genetic variants found with three different methods.

Pt_ID	MULTISAMPLES Test	SINGLETON Test	Inborn Errors of Immunity (IEI) PANEL (538 Genes)
**Pt1**	** MST1R **	MPEG1	CARD10
	CASP8		LAT
**Pt2**	** DNM2 ** / MRE11/IRF7/CD36		
	XRCC2/ ** NBAS **		** TGFBR2 **
	SIGLEC1/MPO/ZNFX1		** RELA ** / TRNT1
**Pt3**	** BRCA1 ** / ** MPO ** ** / ** CASP5		IFNAR1/ ** SLC7A7 **
	EPX ** / ** RIF1		
**Pt4**	** RANBP2 ** /DOCK1/DOCK8/PLCH2/PIEZO1		
	** SPINK5 **		
**Pt5**	EP300/VPS13B/ZNFX1		AP3D1
	** ATP7B ** /PLCL2		
**Pt6**	BCLAF1/EPX/MYO9B/PDGFRB/PIK3C2G	** CASP4 **	ADAM17/IL18BP/IPO8/LRRC32
**Pt7**	FANCM/MRE11		SLCO2A1/NLRP1
**Pt8**	PLCG2		POLD1
	ATP7B/CXCR1/CD36		
**Pt9**	POLE/DOCK8/MAP3K21/SIGLEC1/ ** DNM2 **		HS3ST6
**Pt10**	** SPINK5 ** ** / ** CXCR1/DOCK1/PLCL2		** POLR3A **
	HYOU1/POLE/RANBP2		
**Pt11**	EP300/ORAI1	PSMB11	
	CHD7/PIK3C3/ ** RAD50 **		
**Pt12**	MYO9B/NBAS/PDGFRB/XRCC2		NLRC4
	ORAI1/PIK3C3/ ** CTC1 ** / PEPD	CBLB	
**Pt13**	DOCK1/EPX ** /FANCM ** /HYOU1		TICAM1
**Pt14**	ADAMTS13/ ** PRKCH/ ** ZNFX1	HAX1	CTNNB1
	EP300		
**Pt15**	** CASP5/ ** DOCK8 ** / ** PIEZO1	** GATA2 **	
	PRKCH		
**Pt16**	ARHGAP21/PRKCH	TET2	** PSMG2 ** /SP110
**Pt17**	BRCA1/MRE11/RIF1/ADAMTS13		RFXAP
	** PEPD ** / ** RAD50 **		
**Pt18**	** ABCB6 ** /IRF7/MST1R/PLCG2	AEN ** / ** ATG4D ** /RSF1 **	
	** ALPI/ ** LYST ** / ** PEPD ** /RAD50/ ** RASAL3		
**Pt19**	BRCA1/MAP3K21/RIF1		ARPC1B/C6
	VPS13B/RASAL3/CTC1/MYOF		
**Pt20**	CASP8/DOCK8/SIGLEC1	**GJB4/** OVCA2 ** / ** ZYX	** MEFV/ ** PLG
**Pt21**	** ATP7B ** / ** N4BP2L2 ** /PEPD / ** RAD50 **		** DCLRE1C **
	** MST1R **		RTEL1
**Pt22**	EP300/MSH4	ACTN1 ** /G6PC3 **	FANCC
	PLCH2/ALPI	TCF3/PGD	
	N4BP2L2	STEAP3	
**Pt23**	PIK3C2G/MYOF	** ITGB3 ** /IGSF6/MSRA/ ** SERPINB3 **	** EXTL3 ** /POLA1
**Pt24**	PDGFRB/ARHGAP21	LTB	NFAT5/TLN1
	MSH4/LYST		
**Pt25**	BCLAF1/PIEZO1	SAMD9	** CARD11 **
	CHD7/ ** PEPD **	CXCR4/RAD23A	

In red: P/LP/hot VUS homozygous or compound heterozygosity in AR transmitted diseases or heterozygous in AD transmitted diseases. In orange: P/LP/hot VUS variants (i) in heterozygosity in AR transmitted disease or (ii) in gene of unknown inheritance or (iii) that would deserve attention due to their role in cancer predisposition, DNA repair or telomer disorders or other disorders (i.e., bleeding). Light grey: cold VUS.

## Data Availability

The original contributions presented in this study are included in the article/Appendix A. Further inquiries can be directed to the corresponding author(s).

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
