# Peer review of "Genetic Landscape of Non-Remitting Neutropenia in Children and Chronic Idiopathic Neutropenia in Adults"

_ijms, 2025, doi:10.3390/ijms26146929_

Round 1

Reviewer 1 Report

Comments and Suggestions for Authors

The authors examined genetic factors and related variables of non-remitting neutropenia in children and chronic idiopathic neutropenia in adults. Although the sample size is limited, this study offers significant insights and is anticipated to enhance future research endeavors. The following are the comments from the reviewer.

Major comments

The health control group reportedly comprised instances of mild COVID-19. However, given that leukopenia is well recognized in COVID-19, this cohort may not serve as a suitable control group for this investigation.

Was the correlation between two or more genetic variants investigated in this study? For instance, Pt#20 possesses MEFV and GJB4; however, do these exhibit any synergistic or additive relationship? Furthermore, how do the authors evaluate cases in which no significant gene variants were identified?

Please include a comprehensive account of the study's limitations and the prospective avenues for future research.

Minor comments

Table 1: The definitions of the abbreviations are insufficient. Please provide definitions for any abbreviations in the footnotes. Please remove the term "number" from the table. Antibodies targeting neutrophils are denoted as “Ab agN” in the footnotes, although this is not reflected in the table. Please specify the patients classified as pediatric.

It was noted that two patients tested positive for anti-neutrophil antibodies; however, which specific cases are referenced in Table 1?

Table 1 enumerates only four instances of “fatigue”; however, the authors assert its prevalence in most cases, presenting a contradiction.

Please provide the definition of the acronym "IEI" upon its initial mention.

Does Figure 2 include numerical units for the horizontal axis?

Author Response

ANSWER TO REV 1

The authors examined genetic factors and related variables of non-remitting neutropenia in children and chronic idiopathic neutropenia in adults. Although the sample size is limited, this study offers significant insights and is anticipated to enhance future research endeavors. The following are the comments from the reviewer.

Major comments

The health control group reportedly comprised instances of mild COVID-19. However, given that leukopenia is well recognized in COVID-19, this cohort may not serve as a suitable control group for this investigation.

Thank you for this important observation. Actually all the healthy controls performed a blood count and they were not either leukopenic or neutropenic .This information was also unambiguosly  added to the manuscript. (line 416-418)

Was the correlation between two or more genetic variants investigated in this study? For instance, Pt#20 possesses MEFV and GJB4; however, do these exhibit any synergistic or additive relationship?

This is a very interesting comment ; indeed synergy could be one of the mechanisms that  contribute  to the phenotype The association of the  variants shown in  the scheme below ( orange and red) was according to  String data base  (https://string-db.org/) with a single link between RAD50 and DCLRE1C genes reported in PT21( Co-Expression, Experimental/Biochemical Data and Association in Curated Database)

PT_ID

ASSOCIATED VARIANTS

PT2

DNM2/ NBAS/TGFBR2/ RELA

PT3

BRCA1/MPO/ SLC7A7

PT4

RANBP2/ SPINK5

PT10

SPINK5/ POLR3A

PT15

CASP5/ GATA2

PT17

PEPD/RAD50

PT18

ABCB6/RSF1/ ALPI/ RAD50

PT20

GJB4/ MEFV

PT21

ATP7B/N4BP2L2/ /RAD50 /DCLRE1C/MST1R

PT23

ITGB3/ SERPINB3/EXTL3

PT25

CARD11/PEPD

Furthermore, how do the authors evaluate cases in which no significant gene variants were identified?Patients without significant gene association  were considered negative

Please include a comprehensive account of the study's limitations and the prospective avenues for future research.

Thank you for the comment : we have added a  text  in the section Conclusion as follow ( starting form line 431)

Even with the limits of the present collection mainly related to low numbers of  enrolled patients whose ancestry was not exstensively collected,   the working hypothesis of neutropenia as a result of disimmunity genetically driven, takes shape.     

The genetic landscape of LO/LL and CIN neutropenias is complex, involving a combination of rare and common variants in genes mainly involved in immune regulation.

The present findings suggest that these conditions are not classical monogenic disorders but rather complex diseases resulting from the interplay of multiple genetic and environmental factors.

All the premises are the basis to design the future goals aimed firstly at increase the sample size to find out shared variants and to analyze any possible epigenetic influence. Recognizing specific patterns of immune-dysregulation emerged as neutropenia,  interpreting  complex genetic testing  and choosing any possible target treatment can be challenging, but might definitely change the future of the affected subjects.

Minor comments

Table 1: The definitions of the abbreviations are insufficient. Please provide definitions for any abbreviations in the footnotes. Table 1 has been extensively reviewed, checked and added with more complete footnotes Moreover it  has been  enriched with 3 more columns: ethnicity , Consanguinity and Family History in order to address some issues useful to describe better the  ancestry of the two groups   

Please remove the term "number" from the table. The term”number” has been removed

Antibodies targeting neutrophils are denoted as “Ab agN” in the footnotes, although this is not reflected in the table.  Agree , previously was not well explained ;the terminology has been aligned  

Please specify the patients classified as pediatric. Thank you for your suggestion; in light gray (Table 1) is highlighted the pediatric population as explained  in the legend

It was noted that two patients tested positive for anti-neutrophil antibodies; however, which specific cases are referenced in Table 1?Thank you for this comment  There was a misunderstanding  on how to indicate  the positivity of  Ab AntN between the adults and the pediatric population. Actually 8/9 pediatric patients were autoimmune ( and by definition  had the positivty of AbAN ), but this was not included in the        autoimmunity column.  The  present table  has been checked and the data now are aligned . Even the text has been corrected  ( line 74-78)

Table 1 enumerates only four instances of “fatigue”; however, the authors assert its prevalence in most cases, presenting a contradiction.

I apologize again for the error , as explained  above,  there were some mistake given  due to from previous descriptions of children separated from adults ( indeed fatigue was one symptoms represnetd mostly in pediatric population ). Now  the data have been aligned  and corrected both in the table and in the text   ( line 74-78)

Please provide the definition of the acronym "IEI" upon its initial mention .

Thank you for the comment.. Now  the acronym IEI is defined as Inborn Errors of Immunity. The fisrt time it is mentioned

Does Figure 2 include numerical units for the horizontal axis?  

  They are sorted by p-value ranking. New figure 2 and legend now included to show this clearly (replacing the old Figure 2).

Reviewer 2 Report

Comments and Suggestions for Authors

This manuscript explores the genetic underpinnings of long-lasting/late-onset neutropenia in children and chronic idiopathic neutropenia in adults through whole exome sequencing. The hypothesis that these phenotypes may share an immune-dysregulation-related genetic background is both timely and compelling. I have the following comments:

  1. The "hot" and "cold" terminology may not be fully understood by all readers. consider clarifying this.
  2. Given the genetic focus, ancestry may influence variant interpretation, such as ethnicity, family history, age at onset. If this data is unavailable please add it to the limitations section.
  3. Are there any reasons this specific IEI panel was used? 
  4. minor English issues: Line 17: “idiopatic” → “idiopathic”; Line 21: “reumathological” → “rheumatological”; Line 269: “respecetevely” → “respectively”

Round 2

Reviewer 1 Report

Comments and Suggestions for Authors

The authors have adequately addressed the reviewer's remarks. However, Table 1 is absent from the revised manuscript. Has it been removed? Please review the revised manuscript once again. Upon proper revision and submission of Table 1, the reviewer will endorse its publication.

Author Response

We have added Table 1 to the main text
